# Molecular detection of avian hepatitis E virus (*Orthohepevirus B*) in chickens, ducks, geese, and western capercaillies in Poland

**Magdalena Siedlecka[1], Agata Kublicka[2], Alina Wieliczko[1], Anna Karolina Matczuk[2]\***

**1** Department of Epizootiology with Clinic of Birds and Exotic Animals, Faculty of Veterinary Medicine, Wrocław University of Environmental and Life Sciences, Wrocław, Poland, **2** Department of Pathology, Division of Microbiology, Faculty of Veterinary Medicine, Wrocław University of Environmental and Life Sciences, Wrocław, Poland

\* anna.matczuk@upwr.edu.pl

## Abstract

*Orthohepevirus B*, commonly known as avian hepatitis E virus (aHEV), causes big liver and spleen disease (BLS) or hepatitis-splenomegaly syndrome (HSS) in chickens. BLS is an emerging disease among chicken flocks in several countries around the world. In our previous studies, serology and molecular biology screening revealed that chicken flocks are widely affected by aHEV in Poland. The present study, which was conducted between 2019 and 2020, aimed to investigate the prevalence of aHEV in chicken flocks and other poultry, including ducks, geese, and turkeys. A total of 307 flocks were examined. In addition, 29 samples from captive wild birds (western capercaillies, *Tetrao urogallus*) were analyzed. In all the investigated poultry species, except turkeys, the nucleic acid sequence covering part of the ORF1 gene of the aHEV genome was detected (34/336 samples, 10.1%). The infection rate was found to be the highest in broiler breeder chicken flocks (14/40 samples; 35%). Phylogenetic analysis of partial ORF1 gene, which encodes helicase, revealed that the obtained sequences belonged to genotypes 2 and 4, while one belonged to genotype 3. Genotype 2 was detected for the first time in domestic geese and ducks, and genotype 4 was detected for the first time in Poland. The study demonstrated the presence of aHEV among the investigated western capercaillies, suggesting that this species is susceptible to aHEV infections and biosecurity is therefore required in western capercaillie breeding facilities.

## Introduction

*Orthohepevirus B*, which is also known as avian hepatitis E virus (aHEV), belongs to the *Hepeviridae* family within the genus *Orthohepevirus*. Four species of *Orthohepevirus* have been identified so far (*Orthohepevirus A*, *B*, *C*, *D*). Among them, only *Orthohepevirus B* species include HEV isolates from chickens and wild birds, while the other three species comprise HEV isolates from mammals [1].

**Data Availability Statement:** All relevant data are within the manuscript and its Supporting Information files. The nucleotide sequences have

been deposited in GenBank. There are 34 accession numbers: OK423501- OK423534.

**Funding:** The APC is financed by Wroclaw University of Environmental and Life Sciences.

**Competing interests:** The authors have declared that no competing interests exist.

Avian HEV is a nonenveloped, single-stranded RNA virus. Its genome has a size of 6.6 kbp and consists of three open reading frames (ORFs) and nonencoding regions at 5'- and 3'-end [2]. ORF1 encodes the domains for nonstructural polyproteins, such as RNA helicase, which are involved in replication. The viral capsid protein (ORF2)-encoding gene is situated at the 3'-end of the genome, overlapped by ORF3 encoding a cytoskeleton-associated phosphoprotein [2–4].

Avian HEV causes big liver and spleen disease (BLS) or hepatitis-splenomegaly syndrome (HSS). It is most prevalent among breeder broiler flocks and commercial laying hens. The disease was observed for the first time in the 1980s among chickens in Australia, and later it was reported in the 1990s in North America [3, 5]. In Europe, it was first described in 2004 in Italy, while in Poland it was detected in 2010 [6, 7]. Avian HEV was identified as an etiological agent of HSS and BLS in the years 1999 and 2001, respectively [3, 8]. The pathogenesis of aHEV infections is still unknown because the virus cannot be efficiently propagated in cell culture and replicates only in infected animals [4].

Thus far, four genotypes of *Orthohepevirus B* have been proposed, but the International Committee on Taxonomy of Viruses ICTV is yet to recognize them. The proposed genotypes share approximately 80% nucleotide identity [7, 9]. In Europe, genotypes 2, 3, and 4 have been observed among chicken flocks, while genotype 1 was detected only once from a sample obtained from feral pigeons [10–13]. Additionally to domestic birds, a novel orthohepevirus distantly related to aHEV, was sequenced from wild birds little egret sampled in Hungary [14].

BLS-affected chickens typically present with enlarged livers and spleens and blood-stained fluid in the abdomen. In addition, a significant decrease in egg production (10%–40%) and an elevated mortality rate (1%–4%) were reported [15]. Since 2016, a new disease associated with aHEV infection has been noticed in layer and broiler breeder hens in several provinces in China. Infection caused hepatic rupture hemorrhage syndrome and was associated with high mortality in hens [16].

Both anti-aHEV antibodies and viral RNA were detected in healthy chicken flocks. This indicates that aHEV is capable of causing subclinical infections [11, 17]. Recent study from apparently healthy layer flocks from Nigeria indicate approximately 10% aHEV RNA prevalence, while 75% of the examined flocks show antibodies against aHEV [18, 19]. This study assessed the prevalence of aHEV among Polish flocks. The analyzed bird species included those that were not previously tested in Poland such as geese, ducks, turkeys, and western capercaillies.

## Materials and methods

### Sample collection

For genetic identification of aHEV isolates, liver and spleen samples were collected from different poultry species: laying hens (n = 49), breeder broilers (n = 40), broilers (n = 53), ducks (n = 35), geese (n = 99), and turkeys (n = 31). Sample collection was carried out between 2019 and 2020. The sample obtained from one flock consisted of livers and spleens collected from five birds (pooled sample). The samples originated from flocks with difficult rearing and increased deaths. In the case of four flocks, veterinary practitioners requested a PCR test for aHEV presence, these flocks were indicated in Table 1.

The study included a total of 307 flocks, which were aged 1–77 weeks and procured from different poultry facilities located in Poland. The study covered 11 polish voivodeships: Pomeranian, Lubusz, Greater Poland, Lower Silesia, Masovian, Łódź, Opole, Silesian, Lesser Poland, Lublin and Subcarpathian. Between 2018 and 2019 dead western capercaillies (n = 29) from Poland's State Forest Districts were sent for necropsy and sample collection. The age of these

**Table 1. Description of positive sequences obtained in this study.**

| No. | Isolate | GenBank accession number | Species | Age (weeks) | Year of isolation | Industry-farm | Genotype/cluster |
|---|---|---|---|---|---|---|---|
| 1 | 1/CL/2020/PL | OK423501 | Laying hen | 40 | 2020 | 1-A | 4 |
| 2 | 2/CL/2020/PL | OK423502 | Laying hen | 35 | 2020 | 2 | 2/1 |
| 3 | 3/CL/2020/PL | OK423503 | Laying hen | 55 | 2020 | 2 | 2/1 |
| 4 | 4/CL/2020/PL | OK423504 | Laying hen | 50 | 2020 | 1-B | 2/3 |
| 5 | 5/CL/2020/PL | OK423505 | Laying hen | 24 | 2020 | 19 | 2/1 |
| 6 | 1/BB/2020/PL | OK423506 | Breeder broiler | 31 | 2020 | 3 | 2/1 |
| 7 | 2/BB/2020/PL | OK423507 | Breeder broiler | 28 | 2020 | 4 | 2/1 |
| 8 | 3/BB/2020/PL | OK423508 | Breeder broiler | 44 | 2020 | 5-A | 2/3 |
| 9 | 4/BB/2020/PL* | OK423509 | Breeder broiler | 44 | 2020 | 5-A | 2/3 |
| 10 | 5/BB/2020/PL | OK423510 | Breeder broiler | 45 | 2020 | 6 | 2/2 |
| 11 | 6/BB/2020/PL | OK423511 | Breeder broiler | 40 | 2020 | 7 | 2/2 |
| 12 | 7/BB/2020/PL | OK423512 | Breeder broiler | 60 | 2020 | 7 | 2/3 |
| 13 | 8/BB/2020/PL | OK423513 | Breeder broiler | 30 | 2020 | 8 | 2/3 |
| 14 | 9/BB/2020/PL | OK423514 | Breeder broiler | 48 | 2020 | 6 | 2/2 |
| 15 | 10/BB/2020/PL | OK423515 | Breeder broiler | 12 | 2020 | 9 | 2/1 |
| 16 | 11/BB/2020/PL | OK423516 | Breeder broiler | 77 | 2020 | 10 | 2/3 |
| 17 | 12/BB/2020/PL | OK423517 | Breeder broiler | 42 | 2020 | 11 | 3 |
| 18 | 13/BB/2020/PL | OK423518 | Breeder broiler | 48 | 2020 | 1-G | 2/1 |
| 19 | 1/B/2020/PL | OK423519 | Broiler | 2 | 2020 | 12 | 4 |
| 20 | 2/B/2020/PL | OK423520 | Broiler | 3 | 2020 | 12 | 4 |
| 21 | 3/B/2020/PL | OK423521 | Broiler | 4 | 2020 | 13 | 4 |
| 22 | 4/B/2020/PL | OK423522 | Broiler | 5 | 2020 | 12 | 4 |
| 23 | 5/B/2020/PL | OK423523 | Broiler | 3 | 2020 | 14 | 4 |
| 24 | 6/B/2020/PL | OK423524 | Broiler | 3 | 2020 | 12 | 4 |
| 25 | 1/D/2020/PL | OK423525 | Duck | 6 | 2020 | 15 | 4 |
| 26 | 1/G/2020/PL | OK423526 | Goose | 2 | 2020 | 16 | 2/3 |
| 27 | 1/CL/2019/PL* | OK423527 | Laying hen | 40 | 2019 | 1-D | 2/3 |
| 28 | 2/CL/2019/PL* | OK423528 | Laying hen | 37 | 2019 | 1-F | 4 |
| 29 | 3/CL/2019/PL* | OK423529 | Laying hen | 28 | 2019 | 1-C | 2/3 |
| 30 | 4/CL/2019/PL | OK423530 | Laying hen | 30 | 2019 | 1-E | 4 |
| 31 | 1/BB/2019/PL | OK423531 | Breeder broiler | 35 | 2019 | 5-B | 2/3 |
| 32 | 1/B/2019/PL | OK423532 | Broiler | 6 | 2019 | 17 | 4 |
| 33 | 1/D/2019/PL | OK423533 | Duck | 2 | 2019 | 18 | 2/3 |
| 34 | 1/WC/2018/PL | OK423534 | Western capercaillie | 80 | 2018 | Poland's State Forest | 2/2 |

* Samples obtained from flocks directed to the aHEV PCR test by veterinary practicioners.

birds was between 1 to 80 weeks. In this case, liver and spleen were also pooled for RNA extraction, but the one sample was taken from each bird.

The collected samples were stored at –80˚C for further analysis.

## Ethical approval

The study was conducted on clinical samples, either dead birds, or internal organ samples collected during on-site necropsies, sent by veterinary practitioners. The use of such samples does not require approval from the Local Ethical Committee, according to Polish animal experiment regulations.

## Isolation and reverse transcription of viral RNA

Viral RNA from pooled livers and spleens was extracted using a Total RNA Mini Plus Kit (A&A Biotechnology, Gdynia, Poland). Using the extracted RNA, cDNA was synthesized with a Maxima H Minus First Strand cDNA Synthesis Kit (Thermo Fisher Scientific Inc., Poland), as per the manufacturer's instructions. An external reverse primer Helic R-1 (5′–CCTCRTGGACCGTWATCGACCC–3′) was used for cDNA synthesis [13, 17].

## Amplification of partial helicase gene

For the screening of aHEV-positive samples, the nested polymerase chain reaction (PCR) assay was performed with two degenerate primers sets that targeted the partial helicase gene region in ORF1, as described previously [13, 17]. A previously sequenced aHEV fragment was used as a positive control (GenBank accession number MH636899.1). The size of the obtained PCR product was 386 bp.

The products resulting from second-round PCR were examined on a 2% agarose gel stained with Midori Green DNA Stain (Nippon Genetics Europe GmbH, Düeren, Germany). The amplified products of ORF1 were excised, purified using Gel-Out (A&A Biotechnology, Gdynia, Poland), and directly sequenced in both directions with Sanger's method (Eurofins Genomics Sequencing GmbH, Cologne, Germany) with the use of PCR primers.

## Sequence analysis

The resulting sequences were compared with that of aHEV reference strains from established genotypes and sequences previously published in Poland, using the BLASTn alignment algorithm at the GenBank database (NCBI). Then, the sequences were aligned using ClustalW and Sequence Identity and Similarity (SIAS) online tool [20].

Phylogenetic trees were constructed using the Maximum-likelihood method and the Neighbor-Joining method with 1000 bootstrap replicates and Kimura 2-parameter model using MEGA 7.0 software [21].

The acquired sequences were deposited in the GenBank database under accession numbers OK423501 to OK423534.

## Statistical analysis

Statistical analysis was performed with Prism 9 (GraphPad, USA). Statistical analysis was performed with unpaired t-test with Welch's correction. Statistical analysis of age of geese samples were performed with unpaired t-test. Minimal data set is contained in S1 Data.

## Results

The aHEV genetic material was detected in 34 out of 336 tested samples (10.1%). Infection was found to be the most prevalent in broiler breeder flocks, and detected in 14 out of 40 tested flocks (35%). In commercial laying hens, the aHEV RNA was detected in 9 out of 49 flocks (18.4%), while in chicken broilers, Pekin ducks, and geese flocks there were 7 positives out of 53 (13.2%), 2 out of 35 (5.7%), and 1 out of 99 tested flocks (1%), respectively. One sample was positive from 29 tested western capercaillies from the same captive flock. No viral RNA was detected in the tested turkey flocks. The obtained results, including flock age upon testing, are presented in Table 1.

There were no statistical differences between the average age of aHEV positive flocks and the average-aged tested birds when each production type was analyzed (Table 2).

**Table 2. Average age of infected flocks versus average age of examined flocks.**

| | Average age of infected flocks (weeks) | Average age of examined flocks (weeks) | P-value | Significance |
|---|---|---|---|---|
| Commercial layer | 37.7 | 31.1 | 0.12 | ns |
| Breeder broiler | 41.7 | 35.8 | 0.23 | ns |
| Broiler | 3.7 | 4.1 | 0.50 | ns |
| Duck | 4.0 | 5.0 | 0.7 | ns |
| Goose | 2.0 | 5.7 | 0.38 | ns |
| Turkey | - | 6.2 | - | - |

Statisticasl analysis was performed with unpaired t-test with Welch's correction (ns—not significant).

Phylogenetic analysis based on the partial helicase gene showed that most of the sequences belonged to genotype 2 (n = 22), 11 belonged to putative genotype 4, and only 1 belonged to genotype 3. In the Neighbor-Joining phylogenetic tree, the sequences of genotype 2 formed three separate clusters (Fig 1). Genotype 2 was found to be dominant in all the tested species. However, in broilers, only genotype 4 was identified. Genotype 3 was detected in one of the broiler breeder flocks, while genotype 4 was detected in three flocks of commercial layers and one flock of ducks. The geographical distribution of these sequences with regard to phylogenetic clusters and poultry species is shown in (Fig 2).

An analysis of the sequences obtained in this study revealed a nucleotide sequence identity of 79.35%–100%. Whereas nucleotide sequence identity was between 74,63% and 96,46% when compared to sequences from GenBank (excluding Polish sequences). All the details are provided in the percent identity matrix in S1 Table.

## Discussion

Poultry production is constantly and rapidly expanding in Poland. The country has been the leading producer of poultry meat in the European Union since 2014. After the detection of BLS, there has been an increased awareness in Poland about the diseases that can impede reproduction and are capable of causing mortality and affecting egg laying performance.

In this study, the overall prevalence of aHEV was found to be 10.1% in the analyzed poultry production types and animal species. However, more significant differences in the prevalence of aHEV viral RNA was noted in the broiler breeder flocks (35% flocks tested), followed by commercial laying hens (18.4%) and broiler flocks (13.2%). A recent Chinese study from the years 2018–2019 showed that the prevalence of aHEV RNA in chickens was 7.92%. In the cited study, only samples from the flocks of age 17–20 weeks were tested. In some Chinese provinces, the prevalence was higher (e.g., 13.85% in Hebei province) [23]. The results on aHEV prevalence might be influenced depending on the type of sample tested. In 2009 study from Spain, where random serum samples were tested for the presence of aHEV RNA, only 1,66% of samples were positive. When fecal and serum material from healthy 29 layer and breeder flocks were tested, 3,45% of samples were aHEV RNA positive [11]. Recent study performed on apparently healthy layer flocks from Nigeria indicates approximately 10% aHEV RNA prevalence. In this study the prevalence in layer chicken flocks was 18,4%, which is higher, then prevalence rates in Nigeria [19]. This could be simply due to the type of material studied, as in Nigeria apparently healthy chickens were examined, while in our study we had samples obtained from flocks with diminished performance. In a recent study, we reported that the seroprevalence of aHEV in chicken flocks was much higher in Poland (56.1%) compared to China (35.9%), whereas in southwest Nigeria 75% examined chicken flocks were positive for aHEV antibodies [13, 18, 24].

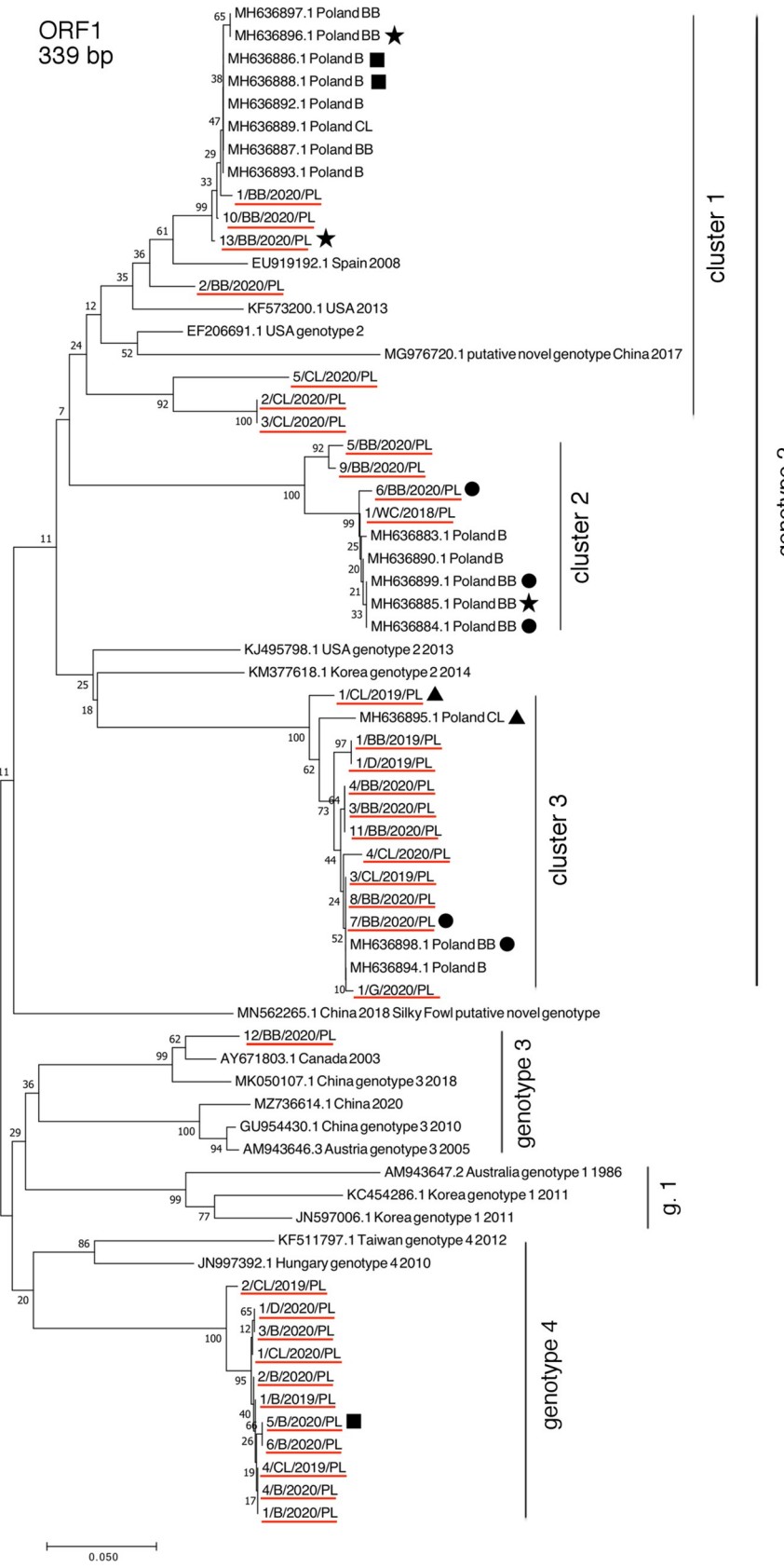

**Fig 1. Phylogenetic analysis based on the partial nucleotide sequences of helicase gene.** Similar sequences and reference genotype sequences were acquired from GenBank database. Sequences obtained in this study are marked with a red line and described in Table 1. The phylogenetic tree was constructed using the Neighbor-Joining algorithm with 1000 bootstrap replicates. Bootstrap values are shown on the tree. CL—laying hen, BB—broiler breeder, B—broiler, D—duck, G—goose, WC—western capercaillie. Symbols: stars, squares, circles and triangles indicate the same farm source of sequence.

The present study indicated that the infection rate in Pekin ducks and geese was minimal, suggesting that these birds are not the main reservoir of aHEV. Data available on the aHEV infection of domestic waterfowl in the literature are limited. Seroprevalence and molecular biology studies have been performed only in chicken flocks and wild birds [4]. In one study, ducks and geese were shown to be infected with aHEV genotype 3. The tested birds were housed in a mixed-type farm along with infected chickens [25].

The occurrence of aHEV has also been demonstrated in free-living birds such as little egret, song thrush, little owl, feral pigeon, and common buzzard [12, 14]. So far, aHEV viral RNA has not been detected in wild *Anseriformes*. Similar to chickens and turkeys, western capercaillies belong to the family of Phasianidae. These birds are protected species in Poland and are bred in three facilities in the country. Some of them are brought back to the forests as part of the reintroduction program. In this study, aHEV viral RNA was detected in one only bird. The sequence belonged to genotype 2 and was very similar to the nucleotide sequence obtained from broiler breeder 6/BB/2020/PL (99.1% identity) and those detected in broiler breeder flocks in Poland in the years 2017 and 2018. The prevalence of aHEV in wild birds has not been investigated in Poland. Few studies available from other countries have demonstrated that aHEV detected in domestic poultry and wild birds are distinct lineages [12, 14]. A high similarity between the sequence obtained from western capercaillie and the sequences from

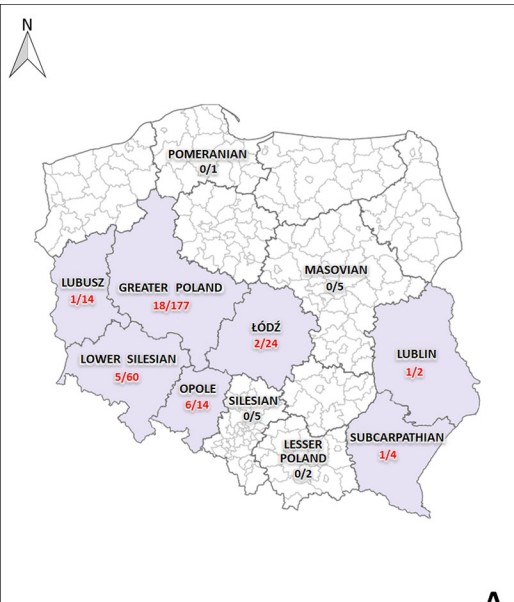
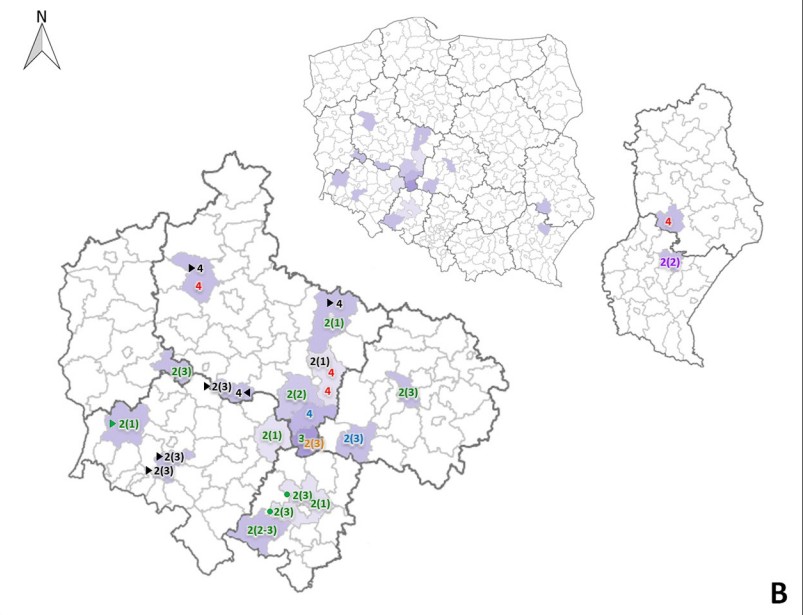

**Fig 2. Geographical distribution and genotype classification of avian HEV viruses.** (A) The location of voivodeships together with the number of infected flocks/number of tested flocks. (B) Poland is divided into voivodeships and counties. Farms belonging to the same industry are marked with a triangle or a circle symbol. Numbers in parentheses are referred to genotype 2 clusters. Commercial laying hen farms marked in black, breeder broiler farms in green, broiler farms in red, duck farms in blue, goose farms in orange and western capercaillies breeding facility in purple. Map reprinted and edited from [22] under a CC BY license, with permission from Wikimedia Commons, original copyright [2020].

broiler breeder chicken is suggestive of indirect transmission, by contaminated feed or water, or caretaker or veterinary staff, and not direct transmission from wild birds.

In this study, aHEV infection was not observed in turkeys. Turkeys were experimentally infected with aHEV [26], but the viral RNA has not been isolated from these species under field conditions. The lack of infection in turkeys can be explained by the fact that the highest bioasecuration standards are followed in turkey farms. Another explanation is that the primers used in this study are not able to detect aHEV circulating in turkey population.

This study involved an analysis of the helicase gene fragment that led to the identification of three different genotypes. Most of the analyzed sequences belonged to genotypes 2 and 4, while only one belonged to genotype 3, which indicates the high genetic diversity of aHEV in Poland. Genotype 3 was detected previously in chickens in Poland during the first aHEV outbreaks [7]. In the present study, only one sequence was found to belong to genotype 3, which shows that this genotype is still present in the field, but not dominant in Poland. This sequence had 96.5% identity to the one detected in Canada in 2003. In a recent study conducted by our group in Poland in 2017–2018, genotype 2 was observed to be the most prevalent [13]. Phylogenetic tree based on partial helicase gene showed that the gene sequences were divided into three clusters, as can be seen in the phylogenetic tree presented in this work (Fig 1). Genotype 4 is described here for the first time in Poland. The sequences belonging to this genotype was 82.0% - 86,7% similar to the sequence reported from Hungary in 2010 which was detected in only one liver sample from BLS outbreaks [27]. Interestingly, this genotype was predominant in one voivodeship (Greater Poland), not in broiler breeder farms but in duck flocks, broiler flocks, and commercial layer flocks.

It is still unknown, whether infections with aHEV are age-associated or not. Conflicting results were described so far and are reviewed in [4]. In our study, there was no statistically significant difference between the age of the tested and infected birds in individual species or chicken production types. We could not statistically analyse all positive versus tested samples, because of the age bias of the flocks from where samples were taken. We had small amounts of samples from young commercial layer flocks or broiler breeder flocks, or samples from older ducks or geese.

In this study we classified genotypes according to analysis performed with the Neighbor-Joining method, because the vast majority of previous publications on aHEV genotyping were performed with this method. However, when the Maximum-likelihood method was used the cluster 3 of genotype 2, seems to be a distinct genotype (S1 Fig.). It should be reminded that thus far, the ICTV does not recognize genotypes of *Orthohepevirus B*. With the emergence of new sequences and genotypes it is urgent to provide proper description of aHEV genotype and appropriate method to distinguish them.

Another interesting observation is that some samples were from the same farms from where we obtained samples and detected sequences in our previous study (Fig 1, S2 Table) [13]. Similar sequences of aHEV partial ORF1 gene were detected from samples collected from those farms, with an exception of the farms indicated with a black square, where sequences were found to belong to genotype 2 in 2017, and to genotype 4 in 2020. This implies that certain aHEV are present on the same farms, which indicates their persistence. Indeed, enteric nonenveloped viruses can resist the action of disinfection agents. Recent data on the virucidal agents effect on nonenveloped *fowl aviadenovirus 1* (FAdV-1), which causes gizzard erosion in chickens, suggest that calcium hydroxide and glutaraldehyde can act as effective disinfection agents at an adequate concentration and ambient temperature [28]. However, there are no data regarding effective virucidal agents against aHEV in field conditions.

## Conclusions

The highest prevalence of aHEV was observed in broiler breeder flocks in Poland. Of the four genotypes of aHEV, genotype 2 has been shown to be widespread so far. Genotype 4, which is newly detected and observed for the first time in Poland, should be investigated in detail with respect to its pathogenic potential in chickens. Natural aHEV infection can be possible in farm ducks, geese and western capercaillies held in captivity. Phylogenetic analysis suggests, that the source of infection is most likely related to indirect transmission from chicken farms. Based on this study, further research should be conducted on aHEV pathogenicity in species other than chickens.

## Supporting information

**S1 Fig. Phylogenetic tree based on maximum-likelihood method.**
(TIF)

**S1 Table. Percent identity matrix of the nucleotides for partial ORF1 among avian HEV strains included in phylogenetic analysis.**
(TIF)

**S2 Table. aHEV isolates collected in the same region in this and previous study from 2017–2018.**
(TIF)

**S1 Data. Minimal data set for statistical analysis performed in Table 2.**
(PDF)

## Acknowledgments

The authors are thankful to Agro-Vet veterinary laboratory staff for technical assistance. They also thank professor Ewa Łukaszewicz (Department of Poultry Breeding, Faculty of Biology and Animal Science, WUELS) for providing western capercaillie samples and Monika Chmielewska-Władyka for providing duck and geese samples.

## Author Contributions

**Conceptualization:** Alina Wieliczko, Anna Karolina Matczuk.

**Data curation:** Anna Karolina Matczuk.

**Formal analysis:** Magdalena Siedlecka, Anna Karolina Matczuk.

**Funding acquisition:** Alina Wieliczko.

**Investigation:** Magdalena Siedlecka, Agata Kublicka, Alina Wieliczko.

**Methodology:** Magdalena Siedlecka, Anna Karolina Matczuk.

**Supervision:** Anna Karolina Matczuk.

**Writing – original draft:** Magdalena Siedlecka, Anna Karolina Matczuk.

**Writing – review & editing:** Alina Wieliczko, Anna Karolina Matczuk.

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
