## [Decision Letter · Decision Letter 0]

12 Jan 2022

PONE-D-21-35714Identification of avian hepevirus (Orthohepevirus B) in chickens, ducks, geese, and western capercaillies in PolandPLOS ONE

Dear Dr. Matczuk,

Thank you for submitting your manuscript to PLOS ONE. After careful consideration, we feel that it has merit but does not fully meet PLOS ONE’s publication criteria as it currently stands. Therefore, we invite you to submit a revised version of the manuscript that addresses the points raised during the review process.

Your manuscript was reviewed by 2 experts in the field. Both identified numerous important problems in your submission. Please review the attached comments and provide point-by-point responses. 

We look forward to receiving your revised manuscript.

Kind regards,

Yury E Khudyakov, PhD

Academic Editor

PLOS ONE

Journal Requirements:

Reviewers' comments:

Reviewer's Responses to Questions

**Comments to the Author**

1. Is the manuscript technically sound, and do the data support the conclusions?

Reviewer #1: Yes

Reviewer #2: Partly

2. Has the statistical analysis been performed appropriately and rigorously? 

Reviewer #1: N/A

Reviewer #2: N/A

3. Have the authors made all data underlying the findings in their manuscript fully available?

Reviewer #1: Yes

Reviewer #2: Yes

4. Is the manuscript presented in an intelligible fashion and written in standard English?

Reviewer #1: Yes

Reviewer #2: Yes

5. Review Comments to the Author

Reviewer #1: SUMMARY OF THE MAIN FINDINGS

In this manuscript, the authors presented the findings of a study designed to investigate the prevalence of avian hepatitis E virus (aHEV) in chicken flocks (broilers, laying hens and broiler breeders), ducks, geese and turkeys aged 1-60 weeks from 307 flocks located in different parts of Poland. Some captive Western capercaillies (Tetrao urogallus) were also included in the study. Using reverse transcription of viral RNA isolated from liver and spleen samples and nested polymerase chain reaction, the authors detected aHEV genetic material in 34 (10.1%) of 336 samples. The highest aHEV infection rate was obtained in broiler breeder flocks (14/40, 35.0%) followed by laying hens (9/49, 18.4%), broilers (7/53, 13.2%), Pekin ducks (2/35, 5.7%) and geese (1/99, 1.0%). Only one (3.5%) of 29 Western capercaillies samples was positive while no viral RNA was detected in the turkey flocks. Phylogenetic analysis based on nucleotide sequences of the partial helicase gene showed that 22 of the identified sequences belonged to genotype 2, 11 to genotype 4, and one to genotype 3 with the genotype 2 sequences forming three distinct clusters. Authors concluded that this is the first detection of aHEV genotype 2 in domestic geese and ducks, and first detection of aHEV genotype 4 in Poland. In addition, they noted that aHEV is most prevalent in broiler breeder flocks in Poland and demonstrated its presence among Western capercaillies, indicating that this avian species is susceptible to the virus.

I believe the manuscript has sufficient quality for publication in PLOS ONE provided the authors can adequately address all the specific comments/issues raised below.

DETAILED REVIEW REPORT

General comments

In my opinion, this manuscript is well written and technically sound. The contents fall within the scope of PLOS ONE and reflect original contribution to knowledge in the field. The methods have been described in sufficient details to allow for reproducibility, while the interpretations are consistent with the objectives of the study and conclusions are justified by the data.

The organization of the article is generally satisfactory although there are some areas where the authors seemed to assume that the reader knows the information they are trying to pass across. For example, the specific locations of the sampling sites in Poland were not given (Lines 75-76), the type of “staff” being referred to in Line 199 was not specified, and what the farms should be “closely monitored” for as well as the host species in which aHEV pathogenicity study should be conducted were not mentioned (Line 239).

In addition, there are some areas where the quality of the English Language needs to be improved. It is advisable that authors engage the services of an English Language editor to improve readability of the text.

Specific comments

Title: I suggest the manuscript title be changed to “Molecular detection of avian hepatitis E virus (Orthohepevirus B) in chickens, ducks, geese, and western capercaillies in Poland”.

Abstract

1. Line 24: It is advisable that authors include the total number of samples here so that readers can have an idea of the total population screened by just reading the Abstract e.g. (34/336 samples, 10.1%).

2. Line 25: Same comments as in 1. above apply here.

Introduction

1. Lines 38, 44, 48 and 233: Authors should endeavor not to start a sentence/paragraph with aHEV. Rather, they should render it as “Avian HEV”.

2. Lines 64-68: Authors have presented results of the study in this introductory section. These should be relocated to the appropriate section in the manuscript.

Materials and methods

1. Lines 71-72: (a) It is important that authors state the voivodeships and counties of Poland from where these samples were collected.

(b) Were these samples from sick or apparently healthy birds? Please specify.

2. Lines 73-74: This period of sample collection is different from the duration stipulated in the Abstract (Lines 19-20). Please reconcile the two.

3. Lines 75-76: The locations of these poultry facilities in Poland should be stated. The map of Poland shown in Figure 2 should have the names of the voivodeships or counties from which samples were collected and those with positive samples.

4. Line 76: Authors should be specific about the internal organ samples being referred to here.

5. Lines 76-77: The age(s) of the Western capercaillies should be stated here.

6. Line 80: Please give examples of these diagnostic procedures to which the samples were subjected.

7. Lines 85-86: Were the Western capercaillies samples from dead or live birds? Please give more information on these birds and how the 29 samples from them were collected.

8. Line 88: Authors should mention the tissues from which viral RNA was extracted.

9. Line 93: Why "first" nested PCR? Did you perform two nested PCR procedures?

Results

1. Line 114: I suggest that authors provide a breakdown of these 34 positive samples into liver and spleen. This might give an indication of the sample that is better suited for aHEV detection.

2. Lines 123-126: The implications of these findings should be highlighted in the Discussion section, otherwise they add no value to the work and should be deleted.

3. Lines 151-152: Supplementary Table S1 is not clear at all. So, it could not be used to confirm the percentage nucleotide identities being referred to in Lines 146-151.

4. Line 159: It would be nice for authors to give the names of locations from where the positive samples were obtained in this map since not all readers are familiar with the different voivodeships and counties in Poland.

Discussion

1. Line 169: I suggest authors retain the description that has been used up until now i.e., aHEV infection instead of BLS. So, this should be changed to "the overall prevalence of aHEV".

2. Line 174: Authors should be consistent with the description of the virus throughout the write-up. You should either retain the use of aHEV or Orthohepevirus B rather than using the two interchangeably.

3. Lines 176-178: I suggest authors reference in their Discussion recently published work such as Osamudiamen et al. (2021) who detected aHEV RNA in tested serum and fecal samples from layer chickens of various ages in Nigeria.

4. Line 206: In my opinion, the sentence will read better if it begins with "Most of the analyzed sequences"

5. Line 217: Authors mentioned in the Results (Line 150-151) that 11 sequences belonged to genotype 4. Is it all the 11 that were 82.2% identical to the Hungarian aHEV sequence or just one of them? Please clarify.

Line 220: I suggest that the sentence starting from “Another” should be the beginning of another paragraph.

Generally, I observed that the Discussion contains too many short paragraphs which reduces the quality of the manuscript. It is advisable that authors merge paragraphs with similar content so that the Discussion can be compact and have a good flow to it. For example, the paragraph that starts on Line 205 should be merged with the one that starts on Line 209.

Lines 227-229: FAdv-1 is a virus and cannot have "virucidal effect". Rather, it is calcium hydroxide and glutaraldehyde that have such effect. So, the sentence should be recast.

Line 231: Replace “virucidals” be with "virucidal agents".

Line 232: I suggest that authors include some few lines on their findings on Western capercaillies in this conclusion section.

Reviewer #2: Matczuk and colleagues present an interesting study on the prevalence of aHEV RNA in different bird species in Poland. The colleagues analysed 336 samples from farmed commercial layer hens, breeder broilers, broilers, ducks, geese, turkeys and also from western capercaillies from Poland’s State Forest Districts.

As a result, the authors detected aHEV RNA in 10.1% of the samples with the highest prevalence in broiler breeders (35%), while the prevalence was low in ducks (5.7%) and geese (1%) and no RNA was detected in turkeys. Also, 1 of 29 samples from western capercaillies was aHEV RNA positive. Most samples belonged to aHEV genotype 2 according to previous studies. For the first time, aHEV genotype 4 has been detected in Poland.

Overall, the study presents new interesting data, shedding more light on the diversity of circulating aHEV strains in Poland. However, there is conflicting information concerning the terms samples/flocks as well as testing strategy (pooled vs. individual testing) raising questions on the resulting prevalence data. Further, phylogenetic analysis based on the Neighbor-joining method is rather outdated and should be performed using the Maximum-likelihood method for a more reliable outcome.

Specific comments

Introduction:

The authors should include the recent reports about potential novel aHEV genotypes and aHEV from wild birds (e.g. Su etal 2018 DOI: 10.1111/tbed.12987 ; Reuter etal 2016 DOI: 10.1016/j.meegid.2016.10.026; Osamudiamen etal 2021 DOI: 10.3390/v13060954).

Lines 63-43: The authors should clarify that the included species were not tested before in Poland, as duck and geese e.g. were tested positive for aHEV in China.

Materials and Methods:

Lines 73-74: In this section, the study was carried out between 2019-2020 while in the abstract it is 2018-2019 (see Abstract, lines 19-20). Please correct.

Lines 73-74: Is there any information on the health status of the birds before sample collection?

Line 74: Were liver and spleen samples pooled? If not, was there any difference in the proportion of positive samples between these 2 tissues?

Line 76: Which internal organ samples? Please give more details.

Line 91: Which primer? Please give the sequence.

Line 101: Were the products sequenced in both directions? Please give this information.

Results:

Line 114: According to the Material and Methods section, the samples were pooled and not tested individually, so how can it be 34 out 336 samples? Or were sample in positive tested pools re-tested individually? Please clarify. Also, “sample” seems to be used synonymous with “flock”. This indicates that each flock is represented by only one sample as nSample = nFlocks in the Materials and Methods section. Please describe more clearly what is meant with “sample” and “flock” and give the number of each collected and tested positive in the Material and Methods and the Results section in a clear way.

Line 117: It must be “18.4%” instead of “18,.%”

Table 1: The average is not correct for all sample types (Laying hens and broilers). Please re-calculate.

Lines 123-126: For broilers the average age of infected vs. uninfected birds is the same. Further, are the results significant? A statistical analysis should be performed.

Figure 1: Please indicate in the legend how long the sequences used for phylogenetic analysis were, as probably the primer sequences were trimmed before analysis?

Figure 1: Sequences of putative novel genotypes which were recently detected in chickens should be included in the phylogenetic analysis. Also, a maximum-likelihood tree is better for phylogenetic analysis as the results are more reliable.

Figure 1: The animal species could be indicated for the sequences from the previously performed study.

Figure 1: For better visibility, the sequences from this study should be marked in bold or otherwise.

Line 146: Please specify which “sequence identity”; nucleotide or amino acid?

Discussion:

Line 169: The chickens are aHEV RNA positive. This does not automatically mean that they have BLS. Or did they show signs of disease?

Lines 173-178: There are also aHEV RNA prevalence data from e.g. Spain or Nigeria (Peralta etal 2009 DOI: 10.1016/j.vetmic.2008.12.010 ; Osamudiamen etal 2021 DOI: 10.3390/v13060954) which should be discussed as it is not clear why for comparison only data from a Chinese study have been chosen. Also, the authors could include their own RNA prevalence data from their previously performed study.

Lines 203-204: It could also mean that the primers which were used are not able to detect aHEV from turkey. Please include this.

Lines 209-210: The first outbreak of what? Please specify.

Line 222: Table S2 is meant.

Conclusion:

Line 233-234: The first sentence seems to be incomplete. aHEV is the most “what” in Poland? Do you mean that aHEV is highly prevalent?

Table S2: The sequences from this study should rather be named by the isolates name and the accession number maybe in brackets to simplify the comparison of the host species of the old and new samples.

6. PLOS authors have the option to publish the peer review history of their article (what does this mean?). If published, this will include your full peer review and any attached files.

Reviewer #1: No

Reviewer #2: No

---

## [Author Response · Author response to Decision Letter 0]

5 Apr 2022

Dear Reviewers,

The co-authors and I would like to thank you for your precious time as well as for your valuable and kind comments considering our manuscript and further revisions required.

We have been able to incorporate changes to reflect the suggestions provided by the reviewers. All amendments in the manuscript are highlighted. The format of the manuscript has been revised according to the submission guidelines.

Here is a point-by-point response to the reviewers’ comments and concerns:

Response to comments and suggestions of Reviewer#1

General comments

In my opinion, this manuscript is well written and technically sound. The contents fall within the scope of PLOS ONE and reflect original contribution to knowledge in the field. The methods have been described in sufficient details to allow for reproducibility, while the interpretations are consistent with the objectives of the study and conclusions are justified by the data.

The organization of the article is generally satisfactory although there are some areas where the authors seemed to assume that the reader knows the information they are trying to pass across. For example, the specific locations of the sampling sites in Poland were not given (Lines 75-76), the type of “staff” being referred to in Line 199 was not specified, and what the farms should be “closely monitored” for as well as the host species in which aHEV pathogenicity study should be conducted were not mentioned (Line 239).

In addition, there are some areas where the quality of the English Language needs to be improved. It is advisable that authors engage the services of an English Language editor to improve readability of the text.

Answer (Ans): We would like to thank you again for the review. We have revised our manuscript according to the Reviewer’s comments and suggestions (please find all changes within the manuscript highlighted in red font. The orginal manuscript was edited by English Langueage editor, we added the file the certificate of the analysis.

Lines 75, 199, and 239 were corrected. The locations have been specified in the text of the manuscript and marked in the corrected Fig 2.

Specific comments

Title: I suggest the manuscript title be changed to “Molecular detection of avian hepatitis E virus (Orthohepevirus B) in chickens, ducks, geese, and western capercaillies in Poland”.

Ans: We are thankful for this suggestion. The title has been changed according to the Reviewer’s suggestion.

Abstract

1. Line 24: It is advisable that authors include the total number of samples here so that readers can have an idea of the total population screened by just reading the Abstract e.g. (34/336 samples, 10.1%).

2. Line 25: Same comments as in 1. above apply here.

Ans: In the lines 24 and 25 corrections have been made as suggested.

Introduction

1. Lines 38, 44, 48 and 233: Authors should endeavor not to start a sentence/paragraph with aHEV. Rather, they should render it as “Avian HEV”.

Ans: Thank you for the suggestion, the sentences have been corrected.

2. Lines 64-68: Authors have presented results of the study in this introductory section. These should be relocated to the appropriate section in the manuscript.

Ans: This paragraph has been removed to not repeat the information about results in the introduction section.

Materials and methods

1. Lines 71-72: (a) It is important that authors state the voivodeships and counties of Poland from where these samples were collected.

Ans: Additional information about the voivodeships of Poland from where these samples were collected, was added in the text. While the counties were marked on the edited map to make the manuscript more clear. We hope that the Reviewer will agree with this solution.

 (b) Were these samples from sick or apparently healthy birds? Please specify.

Ans: The answer to the Reviewer’s question is included in the modified sentence:

“The samples originated from flocks with difficult rearing and increased deaths. In the case of four flocks, veterinary practitioners requested a PCR test for aHEV presence, these flocks were indicated in Table 1.” 

2. Lines 73-74: This period of sample collection is different from the duration stipulated in the Abstract (Lines 19-20). Please reconcile the two.

Ans: We apologize for this oversight. The amendment was introduced in the text.

3. Lines 75-76: The locations of these poultry facilities in Poland should be stated. The map of Poland shown in Figure 2 should have the names of the voivodeships or counties from which samples were collected and those with positive samples.

Ans: We agree with this comment. Therefore, we added the locations of poultry flocks and edited the map.

4. Line 76: Authors should be specific about the internal organ samples being referred to here.

Ans: Liver and spleen samples were taken from western caperacillies too. This sentence has been modified according to the Reviewer's recommendations.

5. Lines 76-77: The age(s) of the Western capercaillies should be stated here.

Ans: Thank you very much for this comment. We provided information about the age of these birds.

6. Line 80: Please give examples of these diagnostic procedures to which the samples were subjected.

Ans: We thank the Reviewer for pointing this out. The phrase "and subjected them to different diagnostic analyses" has been removed from this sentence

The answer to reviewer question is:

All of the samples were subjected to standard bacteriological examination. Some flocks were subjected to PCR tests to detect: IBD infectious bursal disease virus , MDV Marek’s disease virus, FadV-1 fowl adenovirus 1, Reovirus. 4 flocks were specifically subjected for aHEV PCR detection. They were labelled in table 1. The duck and geese flocks were subjected to qPCR against GPV goose parvovirus that detect classical GPV and variant nGPV, GHPV goose hemorrhagic polyoma virus, some flocks were subjected to PCR for Mycoplasma sp. detection. Western capercilles samples were subjected to bacteriological examination.

However we think that supplementing this information would have no impact on the scientific quality of the manuscript. We hope that the Reviewer will agree with this statement. However, in Table 1 we marked the sequences from flocks that were specifically sent for aHEV PCR.

7. Lines 85-86: Were the Western capercaillies samples from dead or live birds? Please give more information on these birds and how the 29 samples from them were collected.

Ans: We checked again and dead western capercaillies (whole birds) were sent for the necropsy. These necropsies were performed by co-author prof. Alina Wieliczko who collected material for storage.

To accomodate also comments from Reviewer 2, we modified the paragraph accordingly. 

8. Line 88: Authors should mention the tissues from which viral RNA was extracted.

Ans: This missing information was added.

9. Line 93: Why "first" nested PCR? Did you perform two nested PCR procedures?

Ans: The word "first" has been used unnecessarily. We performed one nested PCR procedure.

Results

1. Line 114: I suggest that authors provide a breakdown of these 34 positive samples into liver and spleen. This might give an indication of the sample that is better suited for aHEV detection.

Ans: We thank the Reviewer for this suggestion.

In fact in our previous paper doi: 10.1007/s00705-018-4089-y we analysed RNA positivity in spleen and liver samples.

To sum up that study aHEV was detected twice as often in liver then in spleens. There were some birds 2/18 that had detectable aHEV RNA only in spleen and not in liver. Taking it into account we decided to pool liver and spleen samples.

2. Lines 123-126: The implications of these findings should be highlighted in the Discussion section, otherwise they add no value to the work and should be deleted.

Ans: With suggestion from reviewer 2 we made statistical analysis on this data. There is no statistical difference, Table 2 was changed, and this result was discussed, lines 387-393 of R1 manuscript.

3. Lines 151-152: Supplementary Table S1 is not clear at all. So, it could not be used to confirm the percentage nucleotide identities being referred to in Lines 146-151.

Ans: S1 table resolution was checked. We think that for data availability it is good to have this table included in the supporting information. 

The result on similarity was from BLASTn and it did not cover all the sequences. The result was changed and now reflects the data presented in S1 Table. 

4. Line 159: It would be nice for authors to give the names of locations from where the positive samples were obtained in this map since not all readers are familiar with the different voivodeships and counties in Poland.

Ans: The exact location cannot be given as this is sensitive data. The location of the farm is just to see if there are geographical clusters of positive flocks. Names of voivodeships were added to the Fig. 1 map. Also the figure now contains the information about all the samples tested. We could not put all the negative flocks on the map as it would be unreadable, but we included numbers of samples tested for each voivodeships. We hope that these changes add value to the manuscript.

Discussion

1. Line 169: I suggest authors retain the description that has been used up until now i.e., aHEV infection instead of BLS. So, this should be changed to "the overall prevalence of aHEV".

2. Line 174: Authors should be consistent with the description of the virus throughout the write-up. You should either retain the use of aHEV or Orthohepevirus B rather than using the 

two interchangeably.

Ans: On the indicated lines 169 and 174, corrections have been made according to the Reviewer’s suggestion. We decided to retain aHEV since it is better known and simpler name then Orthohepevirus B.

3. Lines 176-178: I suggest authors reference in their Discussion recently published work such as Osamudiamen et al. (2021) who detected aHEV RNA in tested serum and fecal samples from layer chickens of various ages in Nigeria.

Ans: We are thankful for this hint. We agree with this and have incorporated your suggestion.

4. Line 206: In my opinion, the sentence will read better if it begins with "Most of the analyzed sequences"

Ans: Thank you for the suggestion, the sentence has been corrected.

5. Line 217: Authors mentioned in the Results (Line 150-151) that 11 sequences belonged to genotype 4. Is it all the 11 that were 82.2% identical to the Hungarian aHEV sequence or just one of them? Please clarify.

Ans: Indeed, 82,2 % is the value for just one sequence. The sentence was corrected, and values for every sequence from that cluster was added, which was: 82.5% - 86,7%. Additionally, we added also reference sequences from genbank to supplementary table S1. Therefore one can check the similarity of each sequence. 

Line 220: I suggest that the sentence starting from “Another” should be the beginning of another paragraph.

Ans: In accordance with the recommendations another paragraph was added.

Generally, I observed that the Discussion contains too many short paragraphs which reduces the quality of the manuscript. It is advisable that authors merge paragraphs with similar content so that the Discussion can be compact and have a good flow to it. For example, the paragraph that starts on Line 205 should be merged with the one that starts on Line 209.

Ans: Mentioned paragraphs were merged. Additionally two more paragraphs were merged. There are 2 new paragraphs of discussion regarding phylogenetic method and age of the birds. 

Lines 227-229: FAdv-1 is a virus and cannot have "virucidal effect". Rather, it is calcium hydroxide and glutaraldehyde that have such effect. So, the sentence should be recast.

Ans: Thank you for noticing this mistake. The sentence has been corrected.

Line 231: Replace “virucidals” be with "virucidal agents".

Ans: The word „virucidals” was replaced in the manuscript with „virucidal agents”.

Line 232: I suggest that authors include some few lines on their findings on western capercaillies in this conclusion section.

Ans: Information on findings on western capercaillies was added to conclusion section.

Response to comments and suggestions of Reviewer #2:

Matczuk and colleagues present an interesting study on the prevalence of aHEV RNA in different bird species in Poland. The colleagues analysed 336 samples from farmed commercial layer hens, breeder broilers, broilers, ducks, geese, turkeys and also from western capercaillies from Poland’s State Forest Districts.

As a result, the authors detected aHEV RNA in 10.1% of the samples with the highest prevalence in broiler breeders (35%), while the prevalence was low in ducks (5.7%) and geese (1%) and no RNA was detected in turkeys. Also, 1 of 29 samples from western capercaillies was aHEV RNA positive. Most samples belonged to aHEV genotype 2 according to previous studies. For the first time, aHEV genotype 4 has been detected in Poland.

Overall, the study presents new interesting data, shedding more light on the diversity of circulating aHEV strains in Poland. However, there is conflicting information concerning the terms samples/flocks as well as testing strategy (pooled vs. individual testing) raising questions on the resulting prevalence data. Further, phylogenetic analysis based on the Neighbor-joining method is rather outdated and should be performed using the Maximum-likelihood method for a more reliable outcome.

Answer (Ans): We would like to thank you again for the review. We have revised our manuscript according to the Reviewer’s comments and suggestions (please find all changes within the manuscript highlighted in green font.

General comments: The types of samples (pooled for the poultry and individual for the western capecraillies) as weel as pooling strategy liver+spleens were explained in more detail in the manuscript’s materials and method section. We hope that this section is clearer now. 

Specific comments

Introduction:

The authors should include the recent reports about potential novel aHEV genotypes and aHEV from wild birds (e.g. Su etal 2018 DOI: 10.1111/tbed.12987 ; Reuter etal 2016 DOI: 10.1016/j.meegid.2016.10.026; Osamudiamen etal 2021 DOI: 10.3390/v13060954).

Ans: These publications were included in the Introduction section. However, only Reuter et al. describes aHEV from wild birds.

Lines 63-43: The authors should clarify that the included species were not tested before in Poland, as duck and geese e.g. were tested positive for aHEV in China.

Ans: The sentence was modified; „ in Poland” was added to this sentence.

Materials and Methods:

Lines 73-74: In this section, the study was carried out between 2019-2020 while in the abstract it is 2018-2019 (see Abstract, lines 19-20). Please correct.

Ans: We apologize for this oversight. The sentence has been corrected.

Lines 73-74: Is there any information on the health status of the birds before sample collection?

Ans: We are thankful for this suggestion. All the samples were from flocks with difficulties with production or elevated death rates. This information was included in the manuscript lines: 85-87.

Line 74: Were liver and spleen samples pooled? If not, was there any difference in the proportion of positive samples between these 2 tissues?

Ans: In fact in our previous paper doi: 10.1007/s00705-018-4089-y we analysed RNA positivity in spleen and liver samples.

To sum up that study aHEV RNA was detected twice as often in liver then in spleens. There were some birds 2/18 that had detectable aHEV RNA only in spleen and not in liver. Taking it into account we decided to pool liver and spleen samples.

Therefore in this study liver and spleen samples were pooled. Additional sentence with rationale of the sampling processing was added.

Line 76: Which internal organ samples? Please give more details.

Ans: Liver and spleen samples were taken from western caperacillies too. This sentence has been modified according to the Reviewer's recommendations.

Line 91: Which primer? Please give the sequence.

Ans: Thank you for pointing out the missing primer sequence, it has been added to the manuscript.

Line 101: Were the products sequenced in both directions? Please give this information.

Ans: The answer to the Reviewer’s question is included in the below sentence: “The amplified products of ORF1 were excised, purified using Gel-Out (A&A Biotechnology, Gdynia, Poland), and directly sequenced in both directions with Sanger’s method (Eurofins Genomics Sequencing GmbH, Cologne, Germany) with the use of PCR primers.”

Results:

Line 114: According to the Material and Methods section, the samples were pooled and not tested individually, so how can it be 34 out 336 samples? Or were sample in positive tested pools re-tested individually? Please clarify. Also, “sample” seems to be used synonymous with “flock”. This indicates that each flock is represented by only one sample as nSample = nFlocks in the Materials and Methods section. Please describe more clearly what is meant with “sample” and “flock” and give the number of each collected and tested positive in the Material and Methods and the Results section in a clear way.

Ans: Indeed the description was misleading due to different naming of farm animal samples versus samples from western capercailles. 

Line 117: It must be “18.4%” instead of “18,.%”

Ans: Thank you for pointing this out. Correction has been made.

Table 1: The average is not correct for all sample types (Laying hens and broilers). Please re-calculate.

Ans: Table 1 and 2 were corrected. There were some mistakes regarding age of the birds that were corrected.

Lines 123-126: For broilers the average age of infected vs. uninfected birds is the same. Further, are the results significant? A statistical analysis should be performed.

Ans: We checked the table again and indeed there was a mistake. The table was corrected. The statistical analysis was performed and is described in the method section and in table legend.

Figure 1: Please indicate in the legend how long the sequences used for phylogenetic analysis were, as probably the primer sequences were trimmed before analysis?

Ans: 339 nucleotide long fragment was analysed. Information was added to the figure.

Figure 1: Sequences of putative novel genotypes which were recently detected in chickens should be included in the phylogenetic analysis. Also, a maximum-likelihood tree is better for phylogenetic analysis as the results are more reliable.

Ans: Only two additional sequences were included in the phylogenetic tree that had a complete genome accessible in GenBank. Unfortunately, other recently detected novel genotypes were either characterised by other fragment of ORF1 gene or of the ORF2 fragment. Those fragments do not cover fragments sequenced in this study, therefore cannot be compared here.

Regarding the phylogenetic method used, we agree that most of the recent publication states that maximum likelihood method is better to discriminate differences in viral sequences. However, vast majority of publications describing genotyping of aHEV was performed with neighbour-joining method, which is scientifically sound algorithm. Therefore the genotypes described and placed in GenBank were analysed with NJ method. We performed the maximum-likelihood analysis and obtained tree was put in Supporting figure S3. There is a difference; cluster 3 of genotype 2 seems to be a distinct genotype. The genotyping method for aHEV should be addressed by bioinformatical analysis, but this analysis is beyond scope of this manuscript.

We discuss on it in the discussion section: lines 376-382.

Figure 1: The animal species could be indicated for the sequences from the previously performed study.

Ans: All of the sequences from previous study are from chickens. The same abbreviations to label type of production was added to the figure, e.g. B – broiler.

Figure 1: For better visibility, the sequences from this study should be marked in bold or otherwise.

Ans: Sequences from this study were underlined with red. Bold was not possible in our Mega version.

Line 146: Please specify which “sequence identity”; nucleotide or amino acid?

Ans: Thank you for this suggestion. We added a missed word “nucleotide”.

Discussion:

Line 169: The chickens are aHEV RNA positive. This does not automatically mean that they have BLS. Or did they show signs of disease?

Ans: We agree with the Reviewer’s suggestion. It was changed to "the overall prevalence of aHEV".

Lines 173-178: There are also aHEV RNA prevalence data from e.g. Spain or Nigeria (Peralta etal 2009 DOI: 10.1016/j.vetmic.2008.12.010 ; Osamudiamen etal 2021 DOI: 10.3390/v13060954) which should be discussed as it is not clear why for comparison only data from a Chinese study have been chosen. Also, the authors could include their own RNA prevalence data from their previously performed study.

Ans: We extended the discussion part on prevalence according to the Reviewer’s suggestion. 

Lines 203-204: It could also mean that the primers which were used are not able to detect aHEV from turkey. Please include this.

Ans: Thank you for this relevant suggestion. We included this information in the manuscript.

„Another explanation is that the primers used in this study are not able to detect aHEV circulating in turkey population.”

Lines 209-210: The first outbreak of what? Please specify.

Ans: Thank you for pointing this out. The phrase “of avian HEV”has been added.

Line 222: Table S2 is meant.

Ans: The correction has been made according to the Reviewer's suggestion.

Conclusion:

Line 233-234: The first sentence seems to be incomplete. aHEV is the most “what” in Poland? Do you mean that aHEV is highly prevalent?

Ans: To avoid ambiguities, according to the Reviewer's suggestion, this sentence was modified to: “The highest prevalence of the aHEV was observed in broiler breeder flocks in Poland”.

Table S2: The sequences from this study should rather be named by the isolates name and the accession number maybe in brackets to simplify the comparison of the host species of the old and new samples.

Ans: The names of the sequences were changed according to the Reviewer's suggestion.

Please let us know if you still have any questions or concerns about the manuscript. 

We will be happy to address them.

Sincerely,

Anna Matczuk

---

## [Editor Report · Decision Letter 1]

30 May 2022

Molecular detection of avian hepatitis E virus (Orthohepevirus B) in chickens, ducks, geese, and western capercaillies in Poland

PONE-D-21-35714R1

Dear Dr. Matczuk,

We’re pleased to inform you that your manuscript has been judged scientifically suitable for publication and will be formally accepted for publication once it meets all outstanding technical requirements.

Kind regards,

Yury E Khudyakov, PhD

Academic Editor

PLOS ONE
---

## [Editor Report · Acceptance letter]

14 Jun 2022

PONE-D-21-35714R1 

Molecular detection of avian hepatitis E virus (*Orthohepevirus B*) in chickens, ducks, geese, and western capercaillies in Poland 

Dear Dr. Matczuk:

I'm pleased to inform you that your manuscript has been deemed suitable for publication in PLOS ONE. Congratulations! Your manuscript is now with our production department. 

Kind regards, 

on behalf of

Dr. Yury E Khudyakov 

Academic Editor

PLOS ONE